



# Validation of a daily satellite-derived Antarctic sea ice velocity product: impacts on ice kinematics

Tian R. Tian[1,2], Alexander D. Fraser[2,1], Noriaki Kimura[3], Chen Zhao[2,1], Petra Heil[4,2]

[1]Institute for Marine and Antarctic Studies, University of Tasmania, Hobart, TAS 7001, Australia

[2]Australian Antarctic Program Partnership, Institute for Marine and Antarctic Studies, University of Tasmania, Hobart, TAS 7001, Australia

[3]Atmosphere and Ocean Research Institute, The University of Tokyo, Kashiwa, Japan

[4]Australian Antarctic Division, 203 Channel Highway, Kingston, TAS 7050, Australia

*Correspondence to*: Tian R. Tian (tian.tian@utas.edu.au)

**Abstract.** Antarctic sea ice kinematic plays a crucial role in shaping the polar climate and ecosystems. Satellite passive microwave-derived sea ice motion data have been used widely for studying sea ice motion and deformation processes, and provide daily, global coverage at a relatively low spatial-resolution (in the order of 60×60 km). In the Arctic, several validated data sets of satellite observations are available and used to study sea ice kinematics, but far fewer validation studies exist for the Antarctic. Here, we compare the widely-used passive microwave-derived Antarctic sea ice motion product by Kimura et al. (2013) with buoy-derived velocities, and interpret the effects of satellite observational configuration on the representation of Antarctic sea ice kinematics. We identify two issues in the Kimura et al. (2013) product: (i) errors in two large triangular areas within the eastern Weddell Sea and western Amundsen Sea relating to an error in the input satellite data composite, and (ii) a more subtle error relating to invalid assumptions for the average sensing time of each pixel. Upon rectification of these, performance of the daily composite sea ice motion product is found to be a function of latitude, relating to the number of satellite swaths incorporated (more swaths further south as tracks converge), and the heterogeneity of the underlying satellite signal (brightness temperature here). Daily sea ice motion vectors calculated using ascending- and descending-only satellite tracks (with a true ~24 h time-scale) are compared with the widely-used combined product (ascending and descending tracks combined together, with an inherent ~39 h time-scale). This comparison reveals that kinematic parameters derived from the shorter time-scale velocity datasets are higher in magnitude than the combined dataset, indicating a high degree of sensitivity to observation time-scale. We conclude that the new generation of "swath-to-swath" (S2S) sea ice velocity datasets, encompassing a range of observational time scales, is necessary to advance future research into sea ice kinematics.

## 1 Introduction



Antarctic sea ice plays a central role for Southern Ocean climate and ecosystems. Sea ice is a crucial component in the
climate system due to its role in controlling the transfer of energy, gases and moisture between the polar ocean and
atmosphere (Walsh, 1983), impact on climate variability through the ice-albedo feedback (Ebert and Curry, 1993), and deep
water formation as well as fresh water budget by modifying the upper ocean salinity and density gradient during ice growth
and melt (Goosse et al., 1997; Yang and Neelin, 1993). The relatively high albedo of sea ice compared to open water,
especially when covered by snow, means that it reflects a large portion of the incoming shortwave radiation, thus strongly
influencing the Earth's surface energy budget as well as the atmospheric thermo-circulation (Curry et al., 1995).
Furthermore, sea ice production and melt modify the near-surface water, e.g., ice production in coastal polynyas may initiate
the formation of dense shelf water, the precursor of Antarctic Bottom Water, one of the densest water masses in the global
circulation (Rintoul et al., 2001). Sea ice also plays an important role in marine ecosystems and biogeochemical cycles, as a
unique habitat for primary production (Arrigo and Thomas, 2004), and is affected by a network of complex chemical,
biological and physical interactions (Dieckmann and Hellmer, 2010). Hence, understanding Antarctic sea ice physical
processes is critical across many disciplines.

Sea ice kinematics describes sea ice motion and deformation. Accurate characterisation of sea ice kinematics is crucial to
investigate modifications of the polar sea ice cover and sea ice mass balance (Kwok, 2011). Sea ice motion is driven by
external oceanic and atmospheric forcing as well as internal stresses (Kottmeier and Sellmann, 1996). It is also impacted by
physical boundary conditions, e.g., coastline, icebergs, or ice tongues, as well as material properties, e.g., thickness,
concentration, and strength (Heil et al., 2011). Sea ice deformation is defined as the spatial gradients in ice velocity driven by
winds, currents, and internal stress (Kirwan, 1975; Heil et al., 1998; Marsan et al., 2004). Sea ice deformation may change
the drag between ocean and air (Spreen et al., 2017), as well as impacting the distribution of ice thickness and strength
(Marsan et al., 2004).

Sea ice kinematics has been studied extensively in the Arctic using satellite observations (e.g., Weiss and Marsan, 2004;
Hutchings et al., 2005; Spreen et al., 2017; Hutter et al., 2018). However, satellite-based research of Antarctic sea ice
kinematics has been much more limited (Emery et al., 1991; Hakkinen, 1995; Heil et al., 1998; Kimura, 2004; Heil et al.,
2009; Giles et al., 2011; Kimura et al., 2013). In order to characterise Antarctic sea ice kinematics and its effects on physical
ice properties accurately, more research on sea ice deformation processes is required (Hutter et al., 2018). With the
development of remote sensing technology, satellite observations of Antarctic sea ice motion have been improved in recent
years (Spreen et al., 2017; Lavergne et al., 2020). In particular, passive microwave (PM)-derived sea ice motion products
give daily, Antarctic-wide coverage with a gridded data format, which may be suitable for characterising large-scale sea ice
kinematics and evaluating basin-scale sea ice simulations (Sumata et al., 2014). However, detailed exploration of this kind of
composite satellite observations is required. The spatial resolution of PM-derived products is relatively low compared to that
provided by *in situ* buoys and synthetic aperture radar (SAR) sea ice motion products, and complete validation of PM-



derived motion datasets has yet to be achieved, owing to coastal contamination, numerous missing data and artifacts, plus limitations in coverage of buoy deployments necessary for complete characterisation and validation (Heil et al., 2006;

Sumata et al., 2014; Szanyi et al., 2016).

The procedure most widely applied for producing daily sea ice motion vectors is the Maximum Cross-Correlation (MCC) algorithm on satellite-derived daily composites of microwave brightness temperature ($T_B$) or backscatter coefficient (Ninnis et al., 1986; Emery et al., 1991; Kimura et al., 2013). The MCC method determines the spatial difference between image

subsets in two consecutive $T_B$ or backscatter coefficient composite maps by maximizing the cross-correlation coefficient between two composite images of $T_B$ or backscatter coefficient. Sea ice motion products derived from daily-averaged maps of the satellite imagery using MCC algorithm have been referred to as "daily-map" (DM) products (Lavergne et al., 2020). DM products typically have a regular nominal time-scale of approximately one or two days, hence potentially bias the estimated sea ice kinematic parameters by not considering a variety of time-scales (Lavergne et al., 2020). Polar orbiting

satellite swath convergence means a number of swaths are merged (typically by per-pixel mean calculation) to produce composite maps of satellite signals ($T_B$ or radar backscatter), which precipitates a reduction in accuracy of sea ice motion vectors (Lavergne et al., 2020). However, due to the advantages of a) regular large-scale coverage with a consistent grid, b) relative ease of calculation, and c) relative ease of use and interpretation for the dataset end user, DM products have been widely used for recent studies of sea ice kinematics (Szanyi et al., 2016; Hutter et al., 2018). Several PM-derived DM

products have been used for investigating sea ice motion in both Arctic and Antarctic, including the Ocean and Sea Ice Satellite Application Facility (OSI SAF) sea ice drift product OSI-405-c (Lavergne, 2016), henceforth referred to as the OSI SAF product; the NASA Polar Pathfinder Daily 25 km Equal-Area Scalable Earth Grid (EASE-Grid) Sea Ice Motion Vectors product (Tschudi et al., 2019) from the National Snow and Ice Data Center (NSIDC), henceforth referred to as the NSIDC product; and the daily sea ice motion product produced by Kimura et al. (2013), henceforth referred to as the

KIMURA product.

Daily input data of the KIMURA product include both the horizontal and vertical polarization channel $T_B$ images for both ascending (ASC) and descending (DES) orbital sections (i.e., four independent images) from the NASA's Advanced Microwave Scanning Radiometer - Earth Observing System (AMSR-E) and Advanced Microwave Scanning Radiometer 2

(AMSR2), on the sun-synchronous Global Change Observation Mission – Water 1 (GCOM-W1) platform. The final KIMURA daily sea ice motion velocity product is then derived as the mean value of these four velocity datasets. AMSR2, used by the KIMURA product after 2012, passes over each point on the Earth surface at approximately the same local (solar) time, and for any given pixel, the sensing times for ASC and DES passes differ. The local overpass time of the ASC pass in the Antarctic sea ice zone (i.e., ~60º to 78º S) is around local 15:00 (Fig.1, red bar), and the DES pass is around local

midnight (Fig. 1, blue bar). At higher latitudes where several adjacent swaths observe the same pixel, the local times of observation for these several swaths are centerd around the nominal overpass time, but offset by several orbital periods (i.e.,





offset from the nominal local time by +/-n×98 minutes, where n can be 1, 2 or more depending on latitude). The daily combined KIMURA sea ice motion product (represented by the green bar in Fig. 1) is the mean of the ASC and DES data. As such, it has a true (nominal) time base of approximately 39 h. This exceeds that of the ASC and DES motion products

(which have a nominal time base of 24 h). Previous buoy-based research demonstrated that shorter time-scale observations can represent sea ice motion and kinematic parameters with higher magnitude than longer time-scale observations (Hoeber and Marianne, 1987), but analogous research for satellite-based observational datasets is quite limited. We postulate that a shorter time-scale sea ice motion dataset (i.e., derived from ASC or DES DM composites, with their inherent ~24 h time-scale) will result in higher estimates of the kinematic parameter magnitude compared to a longer time-scale dataset (i.e., a

product combining ASC and DES data). Thus, to investigate the influence of the time-scale on the accuracy of the PM-derived sea ice motion and derived kinematic parameter magnitude, individual ASC and DES KIMURA datasets are generated and investigated here.

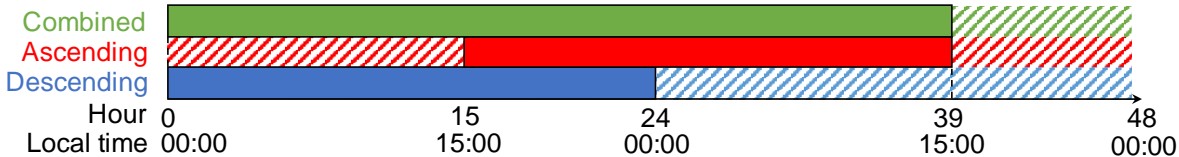

**Figure 1.** Representation of the time-scale of the combined (ASC and DES averaged together) KIMURA daily sea ice motion product, as well as the time-scales of ASC and DES components.

Recently sea ice motion has been estimated from PM-derived $T_B$ partial-overlap swath pairs from a wide range of temporal baselines (from one orbit of ~90 minutes, to 3 days). This approach is referred as "swath-to-swath" (S2S) (Lavergne et al.,

2020). For S2S, the time base for derived motion is a function of the temporal separation of individual swath in each pair, hence differs between pairs (Lavergne et al., 2020). As such, the raw S2S-derived motion swath do not constitute a regular, full coverage time series. However, since S2S most accurately provides ice motion from induvial swaths across a range of time bases, Lavergne et al. (2020) concluded that sea ice motion based on S2S is more accurate than that from DM.

We hypothesise that a greater number of satellite swaths used for averaged maps of satellite signals (e.g., $T_B$ DMs) may negatively impact the performance of PM-derived sea ice motion vectors. Here, we investigate this hypothesis by performing comparisons against drifting buoy-derived motion in different latitude ranges with the expectation that DM-based KIMURA dataset performs better at lower latitudes, where swath overlap averaging is lower. We note that the OSI SAF sea ice motion product is derived by averaging over two days to increase the signal-to-noise ratio of this operational product (Lavergne,

2016). This extended averaging interval arises, in part, due to OSI SAF integrating multiple sensors. Because of the extended time base of the OSI SAF product, direct comparison with the NSIDC and KIMURA datasets in a kinematic parameters context is not possible. Hence the OSI SAF dataset is not considered further in this study, although implications of a longer





temporal baseline are considered in the discussion. Version 3 of NSIDC dataset is known to display a high degree of

autocorrelation in neighbouring vectors (Szanyi et al., 2016) which would be detrimental to the derivation of kinematic

parameters. Version 4 of the NSIDC product has reduced this autocorrelation to some extent (Tschudi et al., 2019), however

this product still assimilates from several instruments, so retrieval of kinematic parameters will suffer from cross-instrument

averaging. Furthermore, as the NSIDC product does not use AMSR2, spatial resolution is inherently lower than KIMURA

dataset. For these reasons, the NSIDC dataset is not considered further here. The KIMURA dataset is the only PM-derived

dataset we consider here; it provides daily gridded sea ice motion vectors covering the complete Southern Ocean, and its 60

km horizontal resolution is suitable for exploring Antarctic sea ice kinematics in this study.

The objectives of this study are to a) evaluate the KIMURA ice motion product separately for the ASC, DES and combined

datasets by comparison with coincident drifting buoy-derived velocities, and b) investigate impacts of the observational

time-scale of the ice motion products on represention of Antarctic sea ice kinematic magnitude. This study will provide

characterisation of the satellite sensor configuration and observational time-scale on the accuracy of derived sea ice motion,

as well as the ability to represent Antarctic sea ice kinematic parameters.

## 2 Datasets

### 2.1 PM-derived sea ice motion

The KIMURA sea ice motion products (Kimura et al., 2013) used here were produced at 60×60 km resolution on a regular

145×145 grid covering the entire Southern Ocean, with daily data from 2012-06-23 to 2020-06-01. These were calculated by

applying the MCC method to 36 GHz, 10 km resolution AMSR2 $T_B$ images (both vertical and horizontal polarization, ASC

and DES). Daily $T_B$ composites, i.e., the input imagery for the MCC algorithm, were obtained from the Arctic Data archive

System (ADS) of the Japanese National Institute of Polar Research (NIPR). The Kimura et al. (2013) dataset combines both

the daily ASC and DES sea ice motion datasets by taking their average. We also produce and analyze daily sea ice motion

vectors computed for the shorter time-scale (24 h) ASC and DES swath composites.

### 2.2 *In situ* buoy-derived sea ice motion

Velocities derived from buoy positions are suitable for validating satellite-derived sea ice motion (Hoeber and Marianne,

1987; Heil and Allison, 1999). Here we evaluate the three KIMURA datasets using three buoys deployed on ice floes in the

Weddell and Ross seas (Table 1 and Fig. 2). These have been named the southern Weddell Sea buoy (Fig. 2, red), the central

Weddell Sea buoy (Fig. 2, green) and the Ross Sea buoy (Fig. 2, orange). In the Weddell Sea, several drifting sea ice buoys

were deployed in 2018 (Schröder, 2018). Only two of these are analyzed in this paper as the buoys were deployed in meso-

scale arrays, hence at sub-grid scale to the KIMURA data. Others in this deployment were near the coast where the





KIMURA ice-motion product suffers from coastal contamination. Raw positions were quality-controlled and any erroneous data removed. A weighted polynomial interpolator (Scargle, 1982; Heil et al., 2001) was applied to the quality-controlled
position measurements to yield an equi-temporal hourly time series. In the Ross Sea, a total of eight position-only buoys were deployed as part of the *Polynyas, Ice Production, and seasonal Evolution in the Ross Sea (PIPERS)* cruise during austral autumn of 2017 (Tison et al., 2020). Four were excluded due to coastal proximity, and another one due to a short time series. Here we analyze only one of the three remaining buoys as they shared similar trajectories.

**Table 1.** Buoy deployment information

| Buoy location | Buoy ID | Latitude Coverage | Active interval |
|---|---|---|---|
| Southern Weddell Sea | ACED-01 | 74.5º S ~ 75.5º S | 2018-04-04 ~ 2018-07-04 |
| Central Weddell Sea | ACED-03 | 71.1º S ~ 75.5º S | 2018-04-08 ~ 2018-08-20 |
| Ross Sea | sbd79220 | 65.0º S ~ 73.4º S | 2017-05-20 ~ 2017-11-26 |

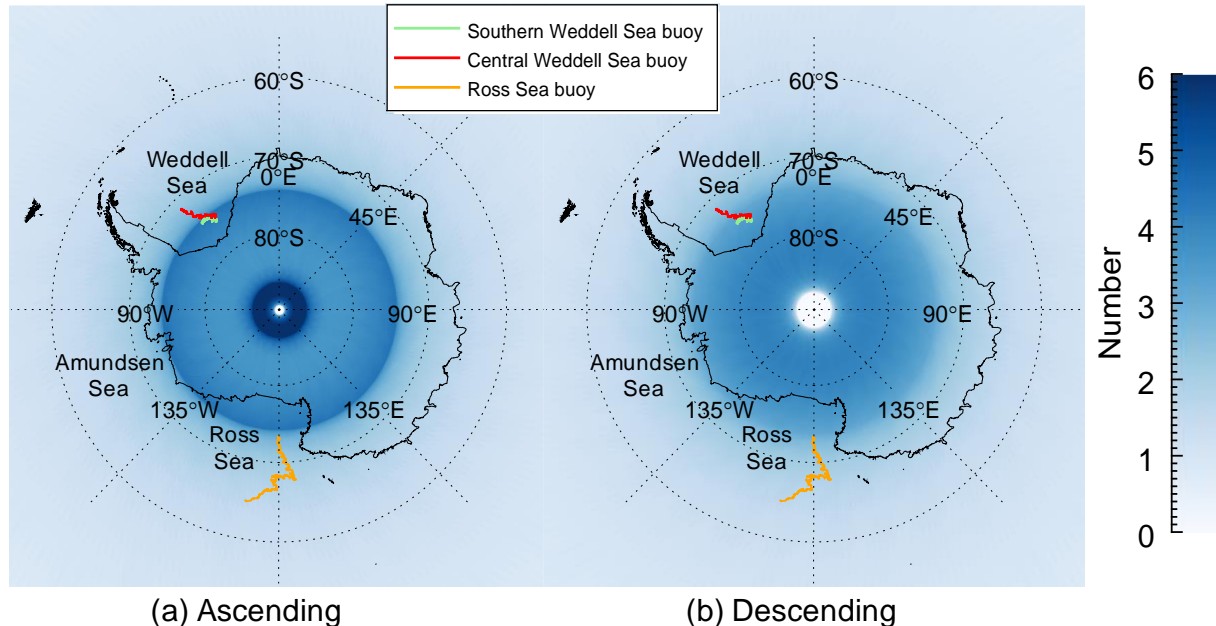


**Figure 2.** Mean daily number of AMSR2 swaths for (a) ASC and (b) DES passes. Trajectories of three buoys use in this study are also shown.

## 3 Methods

### 3.1 Validation of KIMURA-derived ice motions datasets using buoy data



To evaluate the skill of the three KIMURA datasets to represent sea ice motion, we compare these with buoy-derived sea ice velocity, using three validation metrics: Pearson's correlation coefficient, regression slope and RMS deviation (RMSD). Since AMSR2 is in a sun-synchronous orbit, it passes over each point on the Earth at roughly the same local (solar) time. For daily motion comparison buoy time stamps, referenced in UTC time, are converted to local (solar) time, to investigate the start and end time for daily motion comparison. To calculate daily buoy velocity, a 24-hour interval of buoy positions is

selected, to coinciding with the satellite overpass time. The choice of the start time for a 24 h interval is informed by calculating all validation metrics in hourly intervals from -48 h (local midnight of two days before) to +48 h (local midnight of two days later). The ASC, DES and combined start times which are found to maximise the correlation between KIMURA and buoy sea ice velocity were fixed for all analyzes. Spatially, KIMURA pixels contributing to the daily velocity were selected by weighting the velocity in the pixels that the buoy passed through in the preceding 24 h.

We note that the number of AMSR2 swaths (Fig. 2, blue shading) increases rapidly poleward of 74° S for both ASC and DES passes. This is particularly evident in the ASC case, where a strong discontinuity is observed at this latitude, owing to the oblique, forward-looking viewing geometry of the instrument. In both ASC and DES cases, more satellite swaths will be merged together at higher latitude (i.e., a higher degree of temporal "smearing" is expected at higher latitudes, with potential detriment to derived motion products). By extension, fewer swaths will be merged at lower latitudes, (i.e., at northern

extremes, the DM product tends toward the S2S product, with reduced temporal smearing). Based on the results of Lavergne et al (2020), we expect better performance of the KIMURA dataset at lower latitudes for this reason. To investigate the effect of compositing more satellite swaths on the accuracy of the KIMURA dataset, buoy comparisons are separated based on their latitude (Table 1). We separate the central Weddell Sea buoy to four 1° latitude segments and the Ross Sea buoy to four 2° latitude segments to support this analysis (the southern Weddell Sea buoy has insufficient latitudinal range for this

analysis).

### 3.2 Calculation of sea ice divergence

Various differential kinematic parameters (DKPs) are frequently used to represent sea ice kinematics (Molinari and Kirwan, 1975; Hutchings et al., 2012): divergence ($D$), shear deformation rate ($S$, pure shear), normal deformation rate ($N$, stretching) and vorticity ($V$, rotation). Here we use $D$ to investigate a hypothesised increase in DKP magnitude when

considering the shorter (24 h) time scale ASC and DES datasets, compared to the combined dataset. To generate daily maps of $D$ from the KIMURA dataset, $D$ at the center of four neighbouring pixels was calculated as:

$$D = \frac{\partial u}{\partial x} + \frac{\partial v}{\partial y}, \tag{1}$$

where $u$ and $v$ are daily sea ice velocities in the eastward and northward directions, respectively, while $x$ and $y$ represent eastward and northward components. We evaluate the magnitude of $D$ for the combined, ASC and DES datasets by





computing the monthly RMS deviation (hereafter RMSD, i.e., a measure of monthly variability). Variability of $D$ over a period of one month is used here to assess the relative magnitude of the DKPs. Furthermore, based on the latitude hypothesis that more satellite swaths become merged causing more smoothing at higher latitudes, we also expect that $D$ RMSD will decrease from north to south.

## 4 Results

### 4.1 KIMURA data assessment

Here, we validate the three KIMURA sea ice motion datasets using daily velocity from buoys in the Ross and Weddell seas. First, we report some problems discovered in the KIMURA datasets.

### 4.1.1 Problem 1: Triangular regions with persistent low velocity

Initial inspection of the KIMURA datasets reveals the existence of two triangular regions with recurring low velocity (every
second day). These are in the western Amundsen Sea for the ASC dataset (triangular region in Fig. 3a) and the Weddell Sea for the DES dataset (triangular region in Fig. 3c). These problem regions are also evident in the combined product, although their impact is masked (halved) due to the averaging of ASC and DES velocities. Upon investigation, this occurs at an acquisition time of around midnight UTC, where swath duplication has erroneously occurred in the $T_B$ DM product (from NIPR, ADS). Such areas with replicated values in the source $T_B$ images manifest as areas of near-zero velocity in the
KIMURA product. To rectify this problem, we produced a new version of the KIMURA product using 10 km resolution AMSR2 Level-3 36 GHz $T_B$ images from the Japan Aerospace Exploration Agency (JAXA) ASC and DES $T_B$ daily composites which do not exhibit erroneous swath duplication. A new combined product is produced as the mean of these two with the same 60 km horizontal grid resolution. Rectified KIMURA velocity maps (Fig. 3b and d) reveal that this problem has been overcome. We note that the sea ice motion velocity in other regions changes slightly, which is due to small
differences between the NIPR and JAXA $T_B$ DM composites, but investigation is beyond the scope of this work.



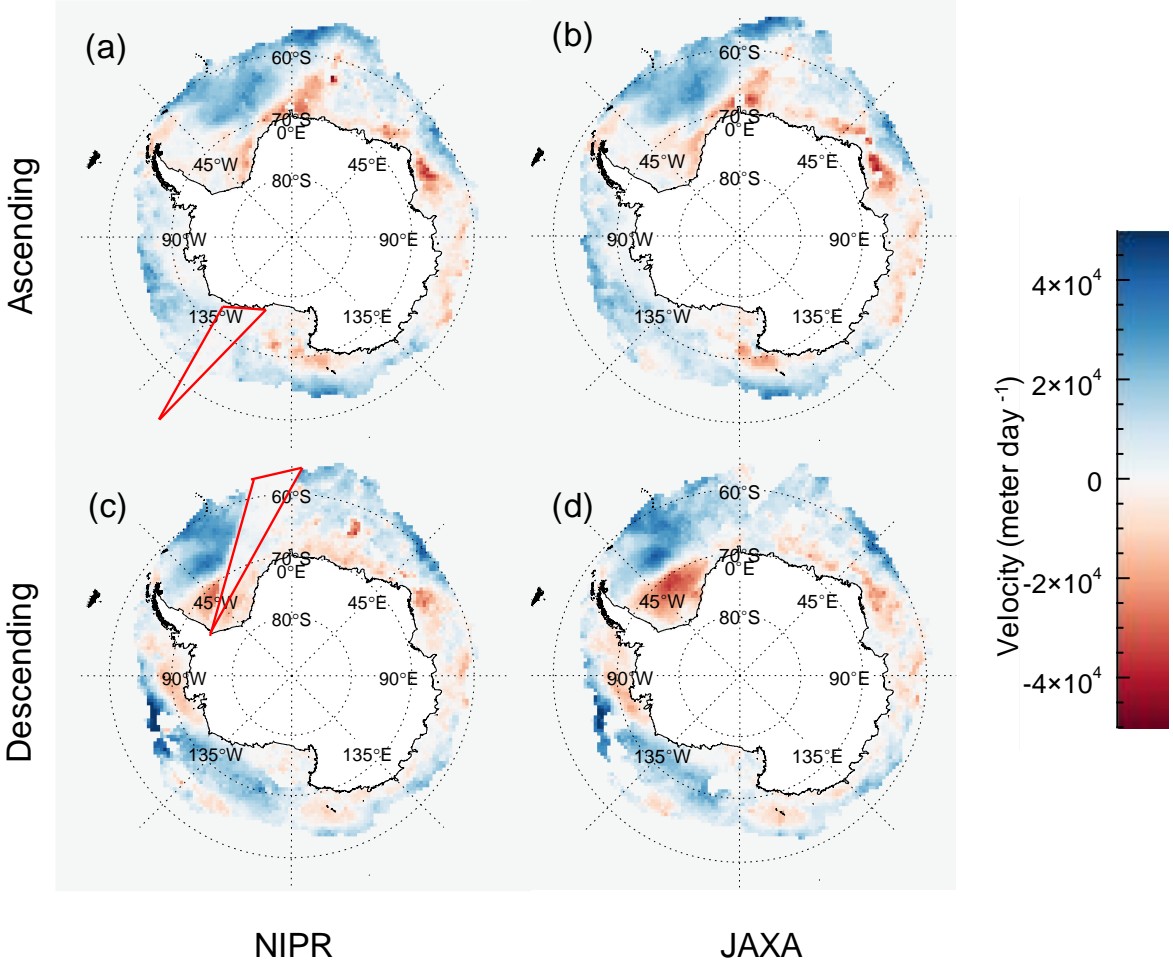

**Figure 3.** Eastward sea ice velocity component of (a) the original KIMURA ASC dataset (produced from NIPR $T_B$ composite), and (b) the rectified KIMURA ASC dataset (produced from JAXA $T_B$ composite) on 2017-09-15. (c) and (d): as for (a) and (b), but for 2017-09-14. The triangular areas shown in red solid lines in (a) and (c) mark the regions of low velocity.

### 4.1.2 Problem 2: Incorrect assumption of 24 h time separation

Our investigation into the source of the first problem revealed a second problem with the KIMURA dataset: an interval of exactly 24 h time separation between consecutive $T_B$ composites was assumed, but rarely correct. We analyze actual time separation between each pair of contiguous days. In order to characterise the variability of time separation, we calculate the standard deviation of time separation for both ASC and DES datasets. The result for September 2017 is displayed in Fig. 4 (all months similar, not shown here).





The mean time separation is close to 24 h throughout each month, however there is a region of high standard deviation around midnight UTC, which occurs in the Amundsen Sea for the ASC swaths (Fig. 4a) and the eastern Weddell Sea for the DES swaths (Fig. 4b). This indicates that the actual time separation of pixels in subsequent daily composites can regularly be up to 34 h, or as low as 14 h in these two areas. Thus, the resulting velocity is regularly incorrect by up to 40%, and the

combined KIMURA dataset exhibits errors for these locations. However, any error is compensated by the following day's velocity field, since the time intervals oscillate around 24 h. As such, it will not have a strong effect on long-term sea ice motion, but is highly detrimental when deriving sea ice kinematic parameters. To fix this problem, the actual time separation needs to be taken into account when producing daily sea ice motion velocity of KIMURA ASC, DES and combined datasets. The appropriate correction has been performed for all remaining analyzes presented here.

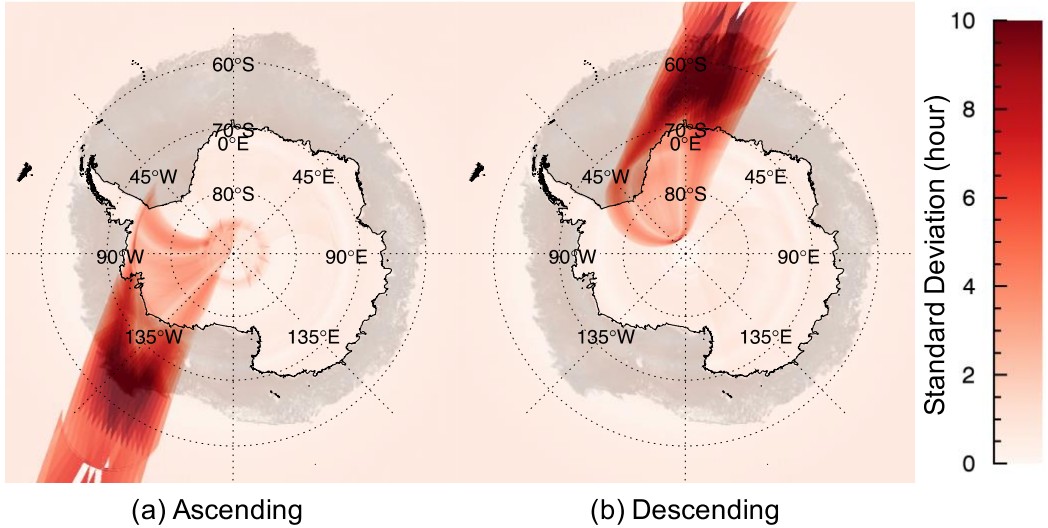


**Figure 4.** Standard deviation of time separation for AMSR2 (a) ASC and (b) DES datasets in September 2017. The shaded background represents AMSR2-derived sea ice concentration on 2017-09-01 (Melsheimer and Spreen, 2019).

In summary, two problems have been discovered in the KIMURA datasets, and these two issues are rectified for the

remainder of this work, as outlined above. Henceforth, the corrected KIMURA products are referred to as the KIMURA$_{new}$ datasets. The official release of the rectified KIMURA sea ice motion is a work in progress.

**4.2 Validation of KIMURA$_{new}$ datasets**

The KIMURA$_{new}$ dataset is compared to buoy-derived sea ice motion. The buoy start and end times needed to maximise correlation coincide exactly with the local (solar) times of corresponding satellite passes, i.e.,15:00, 0:00 and 8:00 UTC for

ASC, DES and combined data, respectively, giving confidence in our methodology. Validation metrics are given in Table 2.





**Table 2.** Comparison between KIMURA_new dataset and buoys

| Validation parameter | Component of velocity | KIMURA_new datasets | Ross Sea buoy (Oct 2017) | Central Weddell Sea buoy (July 2018) | Southern Weddell Sea buoy (May 2018) |
|---|---|---|---|---|---|
| Correlation coefficient | Eastward | ASC | 0.92 | 0.94 | 0.92 |
| | | DES | 0.90 | 0.92 | 0.80 |
| | | Combined | 0.92 | 0.94 | 0.87 |
| | Northward | ASC | 0.94 | 0.97 | 0.97 |
| | | DES | 0.95 | 0.95 | 0.95 |
| | | Combined | 0.96 | 0.95 | 0.96 |
| Regression slope | Eastward | ASC | 0.98 | 0.95 | 0.98 |
| | | DES | 0.97 | 0.87 | 0.99 |
| | | Combined | 1.06 | 0.97 | 1.17 |
| | Northward | ASC | 0.78 | 0.83 | 1.07 |
| | | DES | 0.91 | 0.91 | 1.10 |
| | | Combined | 0.92 | 0.93 | 1.10 |
| RMSD (m day$^{-1}$) | Eastward | ASC | $4.9 \times 10^3$ | $2.1 \times 10^3$ | $2.5 \times 10^3$ |
| | | DES | $5.6 \times 10^3$ | $2.5 \times 10^3$ | $3.7 \times 10^3$ |
| | | Combined | $5.0 \times 10^3$ | $2.0 \times 10^3$ | $3.3 \times 10^3$ |
| | Northward | ASC | $5.8 \times 10^3$ | $2.6 \times 10^3$ | $2.3 \times 10^3$ |
| | | DES | $5.0 \times 10^3$ | $2.7 \times 10^3$ | $3.2 \times 10^3$ |
| | | Combined | $4.4 \times 10^3$ | $2.6 \times 10^3$ | $3.0 \times 10^3$ |

The validation metrics (Table 2) demonstrate that the KIMURA_new datasets represent buoy velocities well. The mean value
of correlation coefficient of comparison between three KIMURA_new datasets and three buoys is 0.93, and of regression slope
is 0.97. We notice that comparison of KIMURA_new with the Ross Sea buoy (Table 2) exhibits relatively large RMSD (4.4 to
5.8 km day$^{-1}$), considerably larger than that of the two Weddell Sea buoys (from 2.0 to 3.7 km day$^{-1}$), as well as larger than
the validation of DM-derived sea ice motion RMSD (from 2.3 to 2.9 km day$^{-1}$) from Lavergne et al. (2020). This difference
in RMSD prompted further investigation into the performance difference between the Ross and Weddell seas. Since the
MCC algorithm requires discernible structure in the underlying $T_B$ imagery to produce accurate vectors, we suspect this
RMSD discrepancy is caused by a differing degree of heterogeneity of $T_B$ between the Ross Sea and Weddell Sea ice during
these buoy deployments. To investigate why the RMSD of the Ross Sea result is higher than that of the Weddell Sea, we





analyze the (spatial) standard deviation of 36 GHz 10 km vertically- and horizontally-polarized T$_B$ (i.e., the input data to the
MCC algorithm) in limited areas around both buoy deployments.


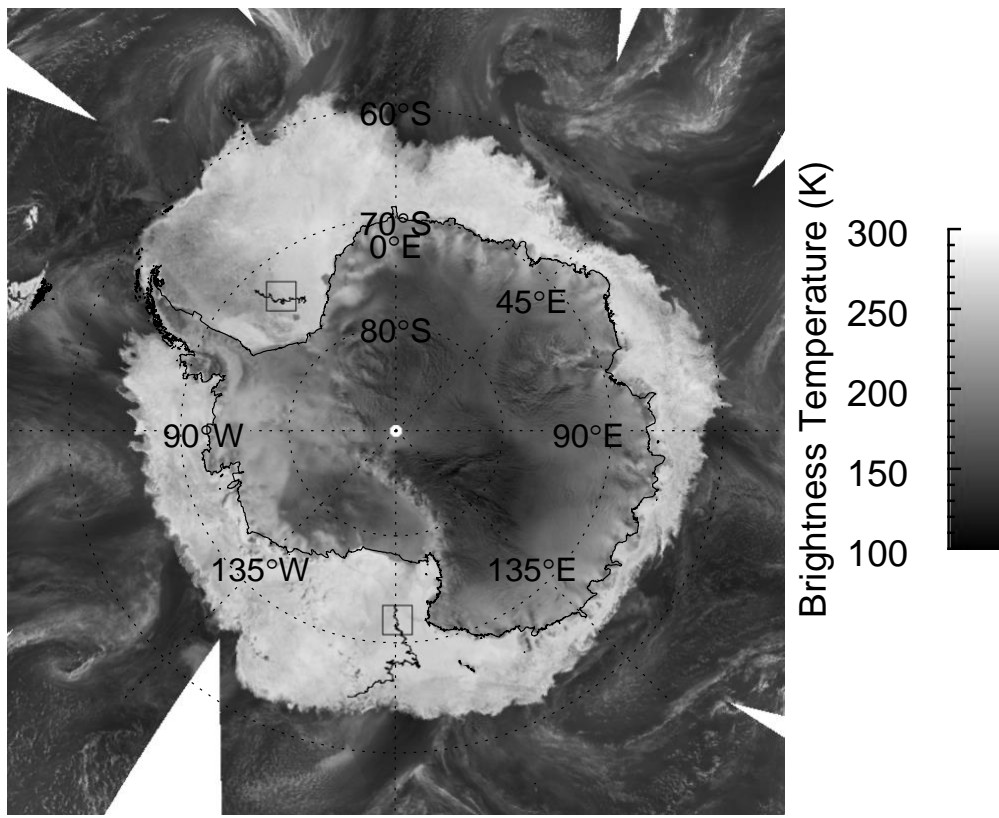

**Figure 5.** ASC, horizontally-polarized T$_B$ composite acquired on 2018-07-01. Buoy trajectories are indicated by black lines in the Weddell
and Ross seas. Square black boxes display the locations of subsets centered around the central Weddell Sea and Ross Sea buoys analyzed
in Table 3.







**Table 3.** Standard deviation of $T_B$ for subregions in the Weddell Sea and Ross Sea shown in Fig. 5. Time periods are the corresponding time of buoys within these subregions. Satellite orbits and polarizations are ASC-horizontal (ASC, H), ASC-vertical (ASC, V), DES-horizontal (DES, H), and DES-vertical (DES, V).

| Region of box | Latitude and longitude of center of box | Time | Orbit and polarization | Std. dev. (K) |
|---|---|---|---|---|
| Weddell Sea | 73.5º S, 40.3º W | 2018-07-01 ~ 2018-07-31 (31 days) | ASC, H | 6.7 |
| | | | ASC, V | 5.4 |
| | | | DES, H | 6.8 |
| | | | DES, V | 5.5 |
| Ross Sea | 72.0º S, 179.0º E | 2017-07-01 ~ 2017-07-31 (31 days) | ASC, H | 5.0 |
| | | | ASC, V | 2.3 |
| | | | DES, H | 4.8 |
| | | | DES, V | 2.1 |

Firstly, we select subregions of 31×31 $T_B$ pixels (i.e., $310 \times 310$ km) that cover the Ross and Weddell Sea buoy areas (Fig. 5). Then we compute the standard deviation within each subregion using all data while the buoys remain within the subregion, and for all four combinations of satellite orbital section and polarization. Standard deviation of $T_B$ in the Ross Sea is lower than the Weddell Sea in all satellite orbital sections and polarizations, with a mean of 3.6 K for Ross Sea and 6.1 K for the Weddell Sea. This indicates that the spatial variability of $T_B$ within the sea ice of the Ross Sea is lower than that of the Weddell Sea (for these cases). The relative homogeneity in the underlying $T_B$ data in the western Ross Sea, a region dominated by new ice formed in the Ross Ice Shelf and Terra Nova Bay polynyas, reduces the ability of the MCC to retrieve accurate sea ice motion, due to less discernible structure in the underlying $T_B$ imagery (in agreement with Meier et al., (2000), who found that errors are maximised in regions of new ice).



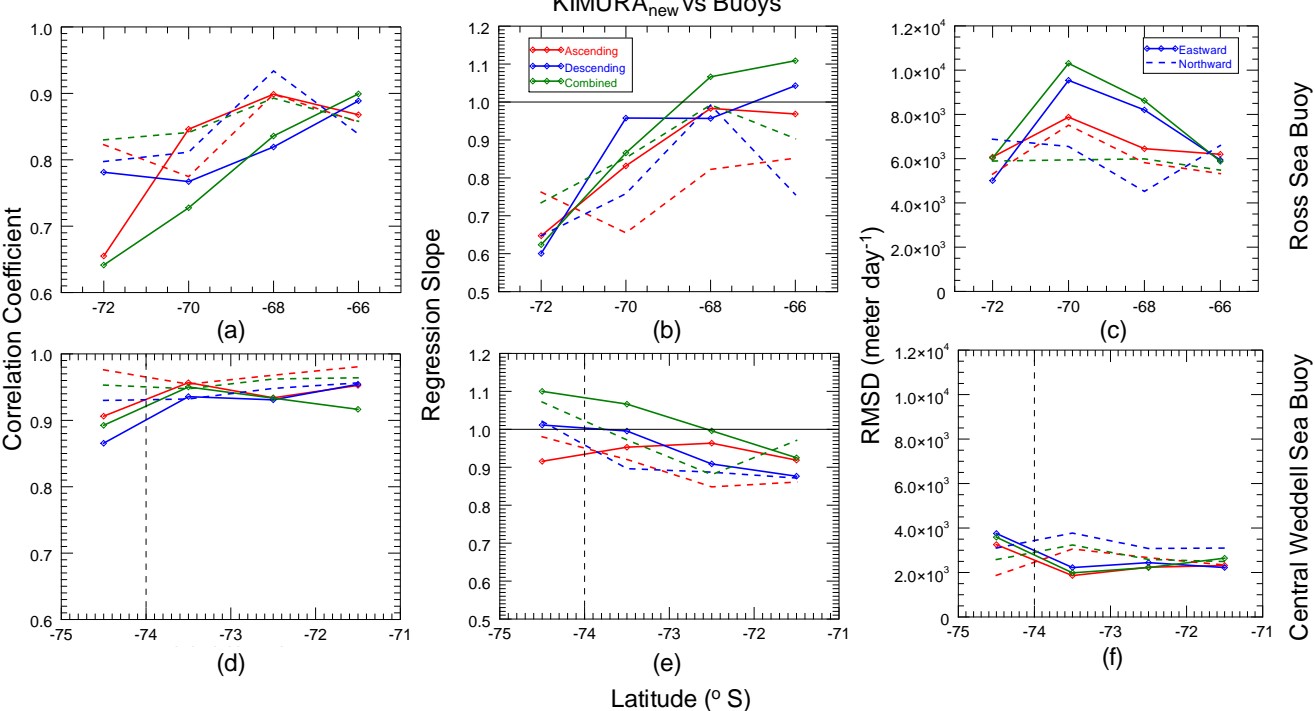

**Figure 6.** Comparison between KIMURA$_{new}$ dataset (ASC-red, DES-blue and combined-green) and buoy velocity using (a, d) correlation
coefficient, (b, e) regression slope and (c, f) RMSD. Vertical dashed lines in the central Weddell Sea panels (lower row) mark the latitude
of the AMSR2 swath discontinuity (74°S) shown in Fig. 2.

To investigate the hypothesis that DM-based daily sea ice motion vector performance is latitude-dependent, we calculate the
three validation metrics in different latitude ranges along the buoy trajectories (Fig. 6). The Southern Weddell Sea buoy is
excluded from this analysis due to insufficient latitudinal range. We find that the validation metrics for both remaining buoys
generally improve as they move further north (Fig. 6). For the central Weddell Sea buoy comparison, eastward velocity
results (Fig. 6 lower row solid lines) improve from south to north across the swath number discontinuity (74°S), confirming
our hypothesis that regions with fewer satellite swaths may yield improved performance for DM-based sea ice motion
retrieval. However, northward ice velocity results in the central Weddell Sea case (dashed lines) exhibit a slight RMSD (Fig.
6f) deterioration from south to north. We attribute this unexpected deterioration to anisotropic T$_B$ heterogeneity, noting that
the heterogeneity in the northward component is lower than that in the eastward component (shown in Table 4), giving
higher confidence in the result of the eastward component, and underscoring the importance of heterogeneous imagery when
using the MCC algorithm.

310





**Table 4.** Eastward and northward components of standard deviation of $T_B$ images in subregion of the central Weddell Sea buoy from July 2018, corresponding to the time buoy resided within the black box (Fig. 5). Satellite orbits and polarizations are ASC-horizontal (ASC, H), ASC-vertical (ASC, V), DES-horizontal (DES, H), and DES-vertical (DES, V).

| Direction | Orbit and polarization | Std. dev. (K) |
|---|---|---|
| Eastward | ASC, H | 3.3 |
| | ASC, V | 2.8 |
| | DES, H | 3.3 |
| | DES, V | 2.7 |
| Northward | ASC, H | 2.7 |
| | ASC, V | 2.3 |
| | DES, H | 2.7 |
| | DES, V | 2.3 |

The validation of the Ross Sea buoy (Fig. 6 upper row) indicates that both the eastward and northward performance improve progressively from south to north. The entire Ross Sea buoy trajectory is located to the north of the swath number discontinuity (i.e., in the relatively low swath number regime, as indicated in Fig. 2), so we suppose that the gradual satellite swath decrease from south to north is not the key reason of increasing accuracy of the KIMURA$_{new}$ product here. Based on our findings about the importance of underlying image heterogeneity, we expect that there may be a higher degree of $T_B$ homogeneity to the south, whereas in the north, approaching the ice edge, lower and patchier sea ice concentration produces relatively high heterogeneity (giving many features for the MCC algorithm to track). In order to represent and analyze the image homogeneity for the southern and northern parts of the Ross Sea buoy trajectory, we examine the standard deviation of $T_B$ images in two 20×20 pixel subsets to approximately match the 2º latitude bin used in computing the validation metrics at the southern and northern ends of its trajectory, respectively (Fig. 7). We find that the standard deviation of $T_B$ in the southern box is much lower than in the northern box, with a mean of 2.3 K for the southern box and 17.0 K for the northern box of the Ross Sea buoy (Table 5). This analysis indicates that $T_B$ in the northern part of Ross Sea buoy trajectory exhibits more structure than in the southern part. Thus, we conclude that the performance of MCC-based sea ice motion algorithms is sensitive to structure in the underlying $T_B$ imagery, with potential regional ramifications which are considered in the discussion.



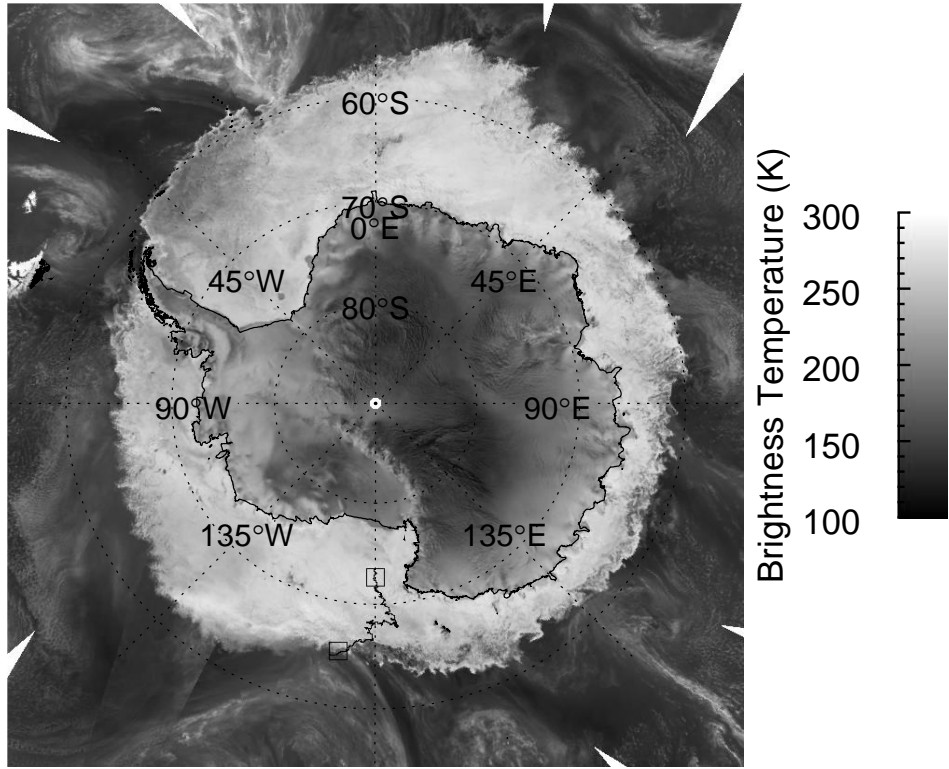

**Figure 7.** Map of ASC-horizontal T$_B$ image on 2017-09-13. Black trajectory and square boxes display subsets at start and end of the Ross Sea buoy.

**Table 5.** Standard deviation of T$_B$ images in two subregions along the Ross Sea buoy trajectory. Times correspond to the buoy within each box. Satellite orbits and polarizations are ASC-horizontal (ASC, H), ASC-vertical (ASC, V), DES-horizontal (DES, H), and DES-vertical (DES, V).

| Latitude of center of box | Time Period | Orbit and polarization | Std. dev. (K) |
|---|---|---|---|
| 73.3 °S | 2017-05-24 ~ | ASC, H | 3.3 |
| | 2017-06-23 | ASC, V | 1.3 |
| | (30 days) | DES, H | 3.3 |
| | | DES, V | 1.3 |
| 65.0 °S | 2017-10-25 ~ | ASC, H | 23.8 |
| | 2017-11-23 | ASC, V | 11.0 |
| | (30 days) | DES, H | 23.2 |
| | | DES, V | 10.5 |





### 4.3 KIMURA$_{new}$-derived sea ice deformation

Accurate sea ice motion data is crucial for calculation of sea ice kinematic parameters. Here we investigate the effect of using the shorter time-scale ASC and DES sea ice motion products to retrieve sea ice $D$, and compare it to those derived from the longer time-scale combined product.

345 #### 4.3.1 Sea ice divergence

Fig. 8 illustrates per-pixel $D$ RMSD for the month of July 2017 derived from the KIMURA$_{new}$ ASC, DES and combined datasets. The spatial mean circumpolar $D$ RMSD (Fig. 8) for the combined dataset ($1.08\times10^{-6}$ s$^{-1}$) is lower in magnitude than that of both the ASC ($1.31\times10^{-6}$ s$^{-1}$) and DES datasets ($1.22\times10^{-6}$ s$^{-1}$), indicating that the ASC and DES datasets can represent a higher magnitude of differential sea ice motion than the combined dataset. The magnitude of derived sea ice kinematics is
350 demonstrated to be sensitive to time-scale, and we have shown that shorter time-scale input data produces higher sea ice $D$ values.

To investigate the impacts of swath number on representing sea ice $D$ magnitude, we compare the July 2017 mean $D$ RMSD calculated by KIMURA$_{new}$ ASC, DES and combined datasets in subregions north and south of the swath number discontinuity (74$^{o}$ S for both ASC and DES, Fig. 2). The mean $D$ RMSD at higher latitudes is about 9%, 14% and 16% lower
355 than at lower latitudes (i.e., north of 74$^{o}$ S) for the ASC, DES and combined dataset, respectively. This indicates that DM-estimated sea ice $D$ RMSD are lower at higher latitudes, potentially due to a greater degree of smoothing due to more merged swath at higher latitudes, i.e., an undesirable bias may be introduced as a consequence of the observation configuration. While we have not investigated physical sea ice properties, which may also give rise to the lower DKP magnitude south of the swath discontinuity, we note that in the major Southern Ocean basins sea ice might be thinner and
360 mechanically weaker at higher southern latitudes than that further north, since much of it has formed recently in coastal polynyas (particularly in the south-western Ross and Weddell seas). As such, we would expect a greater magnitude of sea ice $D$ variability there. In this case, using an S2S dataset to derive sea ice motion may be required to remove observational bias.





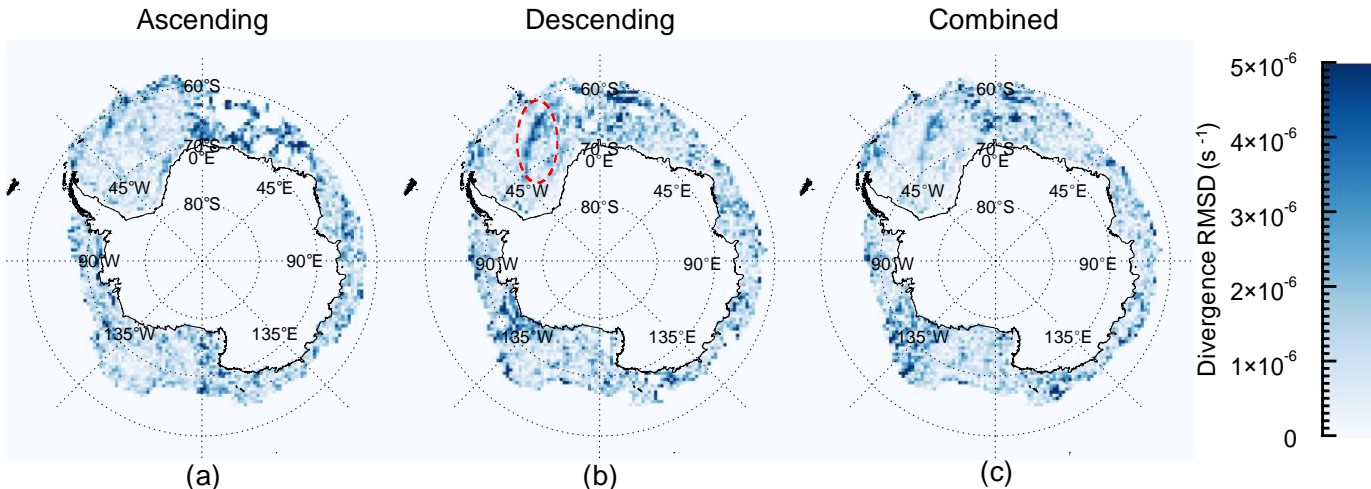

**Figure 8.** Mean $D$ RMSD for (a) ASC, (b) DES and (c) combined KIMURA$_{new}$ datasets of July 2017. The red dashed ellipse highlights the
high $D$ RMSD filamentary structure in the DES map, described in the text.

Fig. 8 also indicates that there are several regions with distinct features when using ASC and DES sea ice motion to calculate
$D$. However, the central northern Weddell Sea stands out, exhibiting a region of high $D$ RMSD in the DES (Fig. 8b, red
dashed circle), which is not mirrored in the ASC RMSD $D$ map. Such filamentary structures exist in the Weddell Sea during
most wintertime months (not shown), and occur in ASC and DES maps equally, but the location is non-stationary. Given the
non-stationary nature of this signal, we suggest that it is not an artifact of the satellite or analysis procedure. Noting the sun-
synchronous observation platform, we suppose that this filamentary structure might be a response to external (oceanic or
atmospheric) forcing with a ~24 h period, such as tidal currents or near-surface wind stress, but further research is beyond
the scope of this paper. That this structure is a) not present in the ASC map, and b) muted in the combined map underscores
the importance of considering observational time-scale and overpass time when interpreting maps of DKPs.

## 5 Discussion

In this study, we firstly analyze DM-based Antarctic sea ice motion retrieval on sea ice kinematics by separating PM-derived
KIMURA$_{new}$ sea ice motion product into ASC and DES. Results show that performance of KIMURA$_{new}$ datasets is a
function of latitude, and that the time-scale of the composite dataset has crucial influence on retrieved sea ice kinematic
magnitude. Here, we discuss the impacts of features of the satellite observational configuration on sea ice kinematics
representation, and how to improve observations in the future.

### 5.1 Impact of features of satellite observational dataset on kinematics representation





We find a strong link between the heterogeneity of underlying satellite-derived maps (i.e., $T_B$ in this study) and the performance of satellite-derived sea ice motion velocity dataset when using MCC algorithm. From the results shown in Table 4, the Weddell Sea $T_B$ composite presents as a relatively heterogeneous $T_B$ field, giving better performance of sea ice
motion velocity vectors compared to the Ross Sea comparison (Sect. 4.2). There is a higher degree of homogeneity in the southern/central Ross Sea than in the northern Ross Sea, and validation metrics (Fig. 6a and b) consequently improve to the north (Sect. 4.2). The KIMURA$_{new}$ product is derived from both horizontal and vertical polarizations, and $T_B$ heterogeneity for horizontal polarization is found here to be generally higher than that for vertical polarization $T_B$ (Table 3, 4 and 5), likely because the difference between emissivity of the first-year sea ice and open ocean at 36 GHz is significantly larger for
horizontal polarization than for vertical polarization (Mathew et al., 2009). Thus, we expect that sea ice motion velocity only derived by horizontal polarization, instead of using both polarizations, may be useful when more accurate characterisation of kinematics is desired.

Higher spatial resolution imagery may also be advantageous in terms of providing underlying imagery which is relatively heterogeneous (i.e., improving MCC algorithm performance). In addition to the results shown in Table 2 (showing a high
RMSD in the Ross Sea buoy validation metrics), we also investigate the standard deviation of $T_B$ in the Ross Sea for several other months in 2017 and 2018 (not shown here), and find that the relatively homogeneous 36 GHz $T_B$ in the Ross Sea is likely to be a persistent feature of ice in this sector. Also, other parameters of satellite signals, such as backscatter may provide more independent structure for the MCC algorithm and may be more successful in a S2S framework than DM. For this reason, the inclusion of scatterometer data may give rise to the good performance of the OSI SAF sea ice drift product
OSI-405 (Lavergne, 2016), albeit giving an (often undesirable) longer time-scale. We also suggest that all MCC-based sea ice motion products report cross-correlation signal-to-noise ratio, as an objective measure of vector confidence (i.e., as a metric for heterogeneity within the MCC search window).

Here we suggest that DM-derived kinematic maps represent not only sea ice kinematics, but also features of the orbit, which is highly undesirable. However, the underlying sea ice motion datasets are still valuable for studies on bulk ice export/flux
(e.g., Drucker et al., 2011), which favour low RMSD datasets assimilating data from many instruments, with temporal baseline a secondary consideration. In this case, the satellite motion datasets based on DM techniques (including the KIMURA dataset) are still valuable (e.g., Drucker et al., 2011). When vector quality is paramount, inclusion of multiple instruments is important. Both S2S and DM products using scatterometer data in addition to radiometers (e.g., the OSI SAF product) would benefit from the inclusion of higher resolution of scatterometer data (e.g., Lindsley and Long, 2010).

A recent study by Lavergne et al. (2020) shows that an S2S sea ice motion dataset provides a diverse set of temporal baselines from which to more rigorously gauge kinematic processes across a wide variety of time-scales. However, interpretation of an S2S product, with its inherently irregular temporal grid and spatial coverage, is likely to be more difficult than DM-derived product. Even when using only a single instrument, DM-derived datasets with their inherently long, low



diversity time-scale cannot track sea ice deformation processes to the same degree as S2S products. Thus, looking to the
future of sea ice kinematic studies, a dataset with the flexibility to provide a wide range of time-scales is necessary.

**5.2 Future development of ice kinematics representation**

To fully and accurately represent sea ice kinematics, some avenues for targeted research pathways are suggested here. From
recent research, S2S-derived sea ice trajectories obtained from AMSR2 imagery have been found to be more accurate than
the DM-derived equivalents (Lavergne et al., 2020). However, S2S products give better coverage of sea ice motion dataset in
the Arctic compared to the Antarctic due to Antarctic sea ice existing at lower latitudes. Thus, combining DM and S2S
concepts together and producing a mixed dataset that takes advantage of both approaches is recommended for improvement
in ice kinematic representation in the future while maintaining ease of use for end users.

More satellites and instruments are needed to acquire more frequent swaths and obtain a more comprehensive sea ice motion
dataset, and to provide a wider diversity of overpass times to avoid the problem of aliasing shown in Fig. 8. For example,
two satellites in a tandem configuration might produce a 12-hour time-scale DM product, rather than the 24-hour time-scale
available from a single platform.

Recently, Lavergne et al. (2020) discussed the Copernicus Imaging Microwave Radiometer (CIMR) mission for both the
Arctic and Antarctica. CIMR will be a conically-scanning sun-synchronous microwave radiometer mission by the European
Space Agency, with a variety of attributes favourable for production of higher accuracy and resolution sea ice motion
products, and is planned for launch in 2026 (Lavergne et al., 2020). However, sun-synchronous platforms are altitude-
constrained (Brown, 1998), which inherently limit their spatial resolution for a given antenna aperture. In the case of CIMR,
there is a need to support other mission objectives, such as sea surface temperature remote sensing, for which sun-
synchronicity is important. Given that sun synchronicity is only important for lower latitude applications (indeed, there is no
diurnal insolation cycle for much of the sea ice season in both hemispheres), we suggest that a future dedicated sea ice PM
radiometer platform may dispense with this constraint, and gain spatial resolution from having a lower orbit.

**6 Conclusions**

This study presents the first comprehensive validation of the KIMURA sea ice motion products (Kimura et al., 2013) for the
Antarctic. Three daily satellite-based PM sea ice motion datasets are investigated: the commonly used combined product
(based on data from both the ASC and DES components of each orbit), as well as the pure ASC and DES components
themselves. Our detailed investigation of the performance of these products reveals two problems with the KIMURA
dataset: triangular regions with persistent low ice motion for the ASC dataset in the western Amundsen Sea and the DES
dataset in the Weddell Sea, and an inaccurate time separation assumption resulting in incorrect daily velocity estimates of up



to 40%. Both problems have been amended for the KIMURA$_{new}$ datasets used in this study, and a corrected version of the
KIMURA dataset for general distribution is under development.

After rectifying both of these problems, we analyze performance of the KIMURA$_{new}$ ASC, DES and combined datasets using
buoy-derived sea ice velocity in the Antarctic. Results indicate that the KIMURA$_{new}$ dataset can represent buoy velocities
well in general, with mean values of correlation coefficient and regression slope of 0.97 and 0.93, respectively. The
validation metrics indicate that at lower latitudes, where fewer AMSR2 swaths are available to be merged to derive the daily
T$_B$ composite image than at higher latitudes, the performance of the KIMURA$_{new}$ datasets is better than at higher latitudes.
Our results support the assertion of Lavergne et al. (2020) that the S2S products probably give more accurate sea ice motion
vectors than the DM-based products. We also report that for the cases studied here, 36 GHz T$_B$ in the Ross Sea is more
homogeneous than in the Weddell Sea, and homogeneity of T$_B$ appears to underlie the reduced performance of the
KIMURA$_{new}$ dataset in the Ross Sea. Lastly, the result of Antarctic sea ice $D$ variability shows that ASC and DES datasets
with their shorter inherent time-scale (~24 h) can represent sea ice deformation processes with higher magnitude than the
longer time-scale (~39 h) combined dataset. We conclude by suggesting that S2S products with their wide diversity of time-
scales (Lavergne et al., 2020) will likely underpin the future of sea ice kinematics research.

*Data availability.* The KIMURA$_{new}$ ASC, DES and combined sea ice motion dataset used in this study is publicly available and can be
accessed using the following DOI: https://data.aad.gov.au/metadata/records/AAS_4506_sea_ice_motion_corrected_Tian_2021

*Author contributions.* TRT processed the sea ice motion data, carried out the analysis and wrote the manuscript. NK provided ASC and
DES data, and contributed to dataset rectification. ADF and PH conceived the project and contributed to the analysis. All authors
contributed to the discussion and provided input during the writing process.

*Competing interests.* The authors declare that they have no conflict of interest.

*Acknowledgements.* AMSR2 Level 3 T$_B$ data and scanning time information were accessed through JAXA's G-Portal. This project
received grant funding from the Australian Government as part of the Antarctic Science Collaboration Initiative program, it contributes to
Project 6 of the Australian Antarctic Program Partnership (Project ID ASCI000002). TRT was supported through an Australian
Government Research Training Program (RTP) scholarship at the Institute for Marine and Antarctic Studies, University of Tasmania. The
Antarctic Science Foundation (ASF) provided generous support and funding to TRT during the COVID-19 pandemic. PH is supported
through the Australian Antarctic Science Project 4506, and the International Space Science Institute (Bern, Switzerland) project #405.
Computation and storage facilities were provided by The National eResearch Collaboration Tools and Resources project (NeCTAR). We
appreciate Drs Damian Murphy and Andrew Klekociuk for technical support and discussions, Dr Misako Kachi for her help on JAXA
scanning time information, and Dr Ted Maksym for the Ross Sea buoy data.



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
