# Peer review of "Rectification and validation of a daily satellite-derived Antarctic sea ice velocity product"

_The Cryosphere, 2021_

## Referee Comment (RC2)

Review for Tian et al., 2021: Validation of a daily satellite-derived Antarctic sea ice velocity product: impacts on ice kinematics:

Overall:  This paper is well-written and presents a detailed analysis to evaluate the accuracy of the Kimura sea ice kinematics product.  Furthermore, the authors go beyond just identifying issues, rather, they improve upon the product and present new results.  This paper serves to assist sea ice motion/kinematic researchers, as well as users of the Kimura product.  I recommend that it is published as is, while addressing a couple comments below.

Table 2 & Figure 6: These are interesting and useful for evaluating the new Kimura product performance.  Perhaps I missed it, but do you summarize the actual ice velocities (buoy and Kimura-derived) anywhere?  It may be worthwhile to examine how the correlation, slope and RMSD perform as a function of velocity.  % differences in speed may decline as the ice moves faster, which typically should be further from the Antarctic coast.

Line 452: "probably" give more accurate sea ice motion?  I'm not sure that's an informational conclusion.  Cite a couple relevant statistics for comparison, then the reader can decide.

---

## Author Response (AR1)

**Author's response**

**RC-1:**

Summary:
Tian and others evaluate the KIMURA sea ice motion algorithm (Kimura et al., 2013) over Antarctic sea ice. They discovered problems with the existing algorithm in the Antarctic and subsequently generated a new dataset. The new dataset was validated use 3 buoy trajectories in Ross and Weddell Seas. Based on the buoy comparison they reported an RMSE between 4.4-5.8 km/day. Overall, this is a useful study/dataset but I think a little more rigor is required. I hope my comments improve the paper.

**Response: We thank the reviewer for their helpful comments. We taken all comments onboard, as detailed below. To clarify we find an RMSE of between 4.4-5.8 km/day only in the Ross Sea. Weddell Sea RMSE are considerably lower than in the Ross Sea, at between 2.0-3.7 km/day, and in line with estimates from Lavergne et al. (2021).**

Major Comment:
I can see the justification for this work in that the previous paper (Kimura et al., 2013) only focused on the Arctic. Further, the improved version corrects problems in the previous version of the data. However, this paper is mostly just a comparison exercise and I am not convinced by the author's justification for not comparing it to the more widely used NSIDC and OSI-SAF sea ice motion products. I think it is important for readers to see how well this new KIMURA dataset compares with these more widely used datasets regardless if the spatiotemporal domains are different. If the other products are not as good it does not hurt to quantitatively show this. A comparison to other datasets will make this manuscript more comprehensive, ensure a larger readership, and encourage the utilization of the new KIMURA sea ice motion dataset.

**Response: We thank the reviewer for their suggestions. We gladly note that during the preparation of this manuscript we compared the NSIDC and OSI SAF velocity products with the same buoys used in this paper. While an important part of the story, we originally deemed the presentation of these findings outside the scope of this manuscript. We agree now that the presentation of this information will add value to the manuscript, and add this velocity comparison (NSIDC, OSI SAF and KIMURA$_{new}$ with buoys) as an appendix (Appendix A), following the format given in Table 2, and also with the scatterplots suggested later in this review. We draw attention to this appendix in section 4.2 of the manuscript.**

Minor Comments:
1. I think the title is a bit miss-leading. The manuscript is about specifically about i) generating new KIMURA sea ice motion dataset and ii) validating it. I think the title needs to be changed to reflect that.

**Response: We thank the reviewer for helping hone the title. We propose the following title which emphasises the generation of the new dataset:**

**Rectification and validation of a daily satellite-derived Antarctic sea ice velocity product.**

2. I found the structure of the paper could be improved with respect to the methods. In Section 2.1, the reader needs more details about how the KIMURA sea ice motion dataset is derived. Following this should the identified problems and then specific details on how they are corrected for the new dataset. In its current form, the methods describing KIMURA sea ice motion dataset lack sufficient detail.

**Response: We thank the reviewer for their comments. While we are generally happy to reference prior methods, we recognise that some more detail here will help the reader understand the techniques without reading the original manuscript. As such, we provide more details on the derivation of the KIMURA sea ice motion in the Datasets (section 2.1).**

**We found that, due to uncovering the dataset issues midway through our study, the structure of this manuscript was difficult to present in a logical order. We decided to include the vector issues in the results section, since this is a major result of our study. We prefer to keep the overall structure as it currently stands, and hope that the addition of more details of the KIMURA dataset derivation in the methods section will help alleviate the reviewer's concerns with the structure. However, we are happy to revisit this decision at the reviewer's insistence.**

3. In Table 2 how many points where used in the comparison? I think a scatterplot needs to be included as a Figure in the manuscript as it is the standard with most sea ice motion comparison studies.

**Response: We have produced scatterplots (Fig. 5) comparing KIMURA and buoy ice-motion data following Table 2. As the data shown in this table are monthly, and the satellite product is daily, there are 30 data points for each comparison scatterplot.**

4. There is no mention of how well the product performs during the melt season. I realize sea ice motion is more challenging in the summer but this needs to at least be mentioned.

**Response: The buoys in this deployment unfortunately demised prior to the summer/melt season. The longest lasting buoy reported until early November, and was far from the ice edge at that time. We include a comment in the manuscript about possible seasonal performance issue of the ice-motion product (at the end of section 4.2), and agree that a follow-up study assessing melt season performance would be interesting.**

5. There are no examples of the new product other than Figure 3. I think perhaps an example with some sea ice motion vectors need to be shown. Perhaps together in a panel with Figure 8?

**Response: We thank the reviewer for suggesting a map of sea ice motion vectors, which will clearly show the improvement of sea ice motion in the new product. Since Figure 8 displays RMSD of divergence, we decided that the ice motion vectors will be better when added to Figure 3. Thus, we add the vectors of KIMURA sea ice motion overlap on the Figure 3. We have also changed the background colour scale into sea ice speed, since eastward and northward components are redundant when displaying vectors. We hope these fully address the reviewer's comment.**

6. When I go online it seems as though data is only available for 2 seasons. Is the complete dataset available? I think it should be.

**Response: Currently our dataset is only available for the years discussed in this manuscript (i.e., covering the buoy deployments). Co-author Noriaki Kimura is working to produce (and release) updates of the corrected data, which will be published soon.**

**RC-2:**
Review for Tian et al., 2021: Validation of a daily satellite-derived Antarctic sea ice velocity product: impacts on ice kinematics: Overall: This paper is well-written and presents a detailed analysis to evaluate the accuracy of the Kimura sea ice kinematics product. Furthermore, the authors go beyond just identifying issues, rather, they improve upon the product and present new results. This paper serves to assist sea ice motion/kinematic researchers, as well as users of the Kimura product. I recommend that it is published as is, while addressing a couple comments below.

**Response: We thank the reviewer for their assessment and recognising the importance of this study.**

Table 2 & Figure 6: These are interesting and useful for evaluating the new Kimura product performance. Perhaps I missed it, but do you summarize the actual ice velocities (buoy and Kimura-derived) anywhere? It may be worthwhile to examine how the correlation, slope and RMSD perform as a function of velocity. % differences in speed may decline as the ice moves faster, which typically should be further from the Antarctic coast.

**Response: We add the actual sea ice velocities of buoy and KIMURA$_{new}$ in this section 4.2 following Table 2. Also, the scatterplots suggested by Reviewer 1 (a comparison between KIMURA$_{new}$ and buoy ice velocity) contribute to addressing the question of a velocity-dependent accuracy.**

Line 452: "probably" give more accurate sea ice motion? I'm not sure that's an informational conclusion. Cite a couple relevant statistics for comparison, then the reader can decide.

**Response: Thank you for picking up on this poor choice of words. We changed the text to "Our work supports the conclusion of Lavergne et al (2021) that S2S data are more accurate than DM, as evidenced by the better buoy validation performance at lower latitudes where fewer swaths are composited together.".**

---

## Author Response (AR2)

**Author's response**

**RC:**

Summary:
The author's have responded to all suggestions adequately.
The only item that I feel still needs clarification (revision) is the buoy comparison in Figure 5. Shouldn't the comparison be for displacement instead of velocity? That is typically how it is done in the literature.

**Response: We thank the reviewer for the suggestion. We have modified the x-axis  label to address this comment (also for Fig. A1).**

The author's could put KIMURA in the title. That is "Rectification and validation of the KIMUARA daily satellite-derived Antarctic sea ice velocity product"

**Response: Our authorship team discussed this point, but prefer that contemporary datasets not be called after an individual. Thus, we prefer to keep the original title: "Rectification and validation of a daily satellite-derived Antarctic sea ice velocity product".**